# Logistic Regression Through the Veil of Imprecise Data

## Abstract

Logistic regression is a popular supervised learning algorithm used to assess the probability of a variable having a binary label based on some predictive features. Standard methods can only deal with precisely known data; however, many datasets have uncertainties that traditional methods either reduce to a single point or completely disregard. This paper shows that it is possible to include these uncertainties by considering an imprecise logistic regression model using the set of possible models obtained from values within the intervals. This method can clearly express the epistemic uncertainty removed by traditional methods.

## 1 Introduction

Logistic regression is used to predict the probability of a binary outcome as a function of some predictive variable, where the time of the event is not essential. In medicine, for example, logistic regression can be used to predict the probability of an individual having a disease where the values of risk factors are known. While logistic regression is most commonly used for binary outcomes, it can be applied to any number of categorical outcomes (Menard, 2010, Chapter 1). However, many decisions and events are binary (yes/no, passed/failed, alive/dead, etc.), and for the sake of simplicity, we will restrict our discussion and examples to binary outcome logistic regression. Additionally, logistic regression, unlike discriminant function analysis, does not require predictor variables to be normally distributed, linearly related or to have equal variance (Press & Wilson, 1978).

There are many practical applications for logistic regression across many different fields. For example, in the medical domain, risk factors in the form of continuous data–such as age–or categorical data–such as gender–may be used to fit a model to predict the probability of a patient surviving an operation (Bagley et al., 2001; Neary et al., 2003). In engineering systems, logistic regression can be used to determine whether a mineshaft is safe (Palei & Das, 2009); to predict the risk of lightning strikes (Lambert & Wheeler, 2005) or landslides (Ohlmacher & Davis, 2003). In the arts, it can be used to explore how education impacts museum attendance or watching a performing arts performance (Kracman, 1996). Within professional sports, it is also possible to predict the probability of game outcomes using logistic regression (Li et al., 2021). Due to its wide range of applications, logistic regression is considered a fundamental machine learning algorithm with many modern programming languages having packages for users to experiment with, such as *Scikit-learn* (Pedregosa et al., 2011) in Python, which has been used for the analysis within this paper.

Traditionally it has been assumed that all of the values of the features and labels used in logistic regression are *precisely* known. This assumption is valid when the sampling uncertainty or natural variability in the data is significant compared to the epistemic uncertainty or if values are missing at random (Ferson et al., 2007). However, in practice, there can be considerable imprecision in both the features and labels used in the regression analysis and the application of the regression model. Analysis using data from combined studies with inconsistent measurement methods can even result in datasets with varying degrees of uncertainty. Likewise, the outcome data can be uncertain if there is ambiguity in the classification scheme (good/bad). However, even relatively straightforward classifications (alive/dead) can yield uncertainty when a subject leaves a study and the outcome is now unknown. Measurement uncertainty is sometimes best represented as intervals, sometimes called "censored data". In the case of continuous variables, the interval reflects the measurement uncertainty, while in the binary outcome, the interval is the vacuous $[0, 1]$ because the correct classification is unclear.

There are multiple methods of dealing with interval data with the features of a logistic regression model, but they often require an approximation that allows the interval to be represented as a single value to be made. Simplifying the process by allowing the use of standard logistic regression techniques. One approach is to treat interval data as uniform distribution (Bertrand, 2000; Billard & Diday, 2000; Bock & Diday, 2001; De Souza et al., 2011) based on the equidistribution hypothesis (Bertrand, 2000) that each possible value can considered to be equally likely. This idea has its roots in the principle of insufficient reason, first described by both Bernoulli and Laplace, and more recently known as the principle of indifference (Keynes, 1921). Alternatively, the interval is commonly represented by the interval's midpoint, which represents the mean and median of a uniform distribution or a random value from within the interval (Osler et al., 2010). While these approaches are computationally expedient, they underrepresent the imprecision by presenting a single middle-of-the-road logistic regression.

Similar methods include performing a conjoint logistic regression using the interval endpoints or averaging separate regressions performed on the endpoints of the intervals (De Souza et al., 2011; 2008). A more general approach is to construct a likelihood function for an interval datum as the difference in cumulative distribution functions of each endpoint (Escobar & Meeker Jr, 1992). While these various methods make different assumptions about the data within the interval ranges, ultimately, they still transform interval data such that the final results can be represented by a single binary logistic regression (De Souza et al., 2011). The use of least squares regression has also been used with interval data (Gioia et al., 2005; Fagundes et al., 2013), but primarily for linear regression models.

The approach proposed within this paper for dealing with interval data in logistic regressions is based on imprecise probabilities and considers the set of models rather than a single one (Walley, 1991; Manski, 2003; Ferson et al., 2007; Nguyen et al., 2012). This is similar to approaches proposed by Utkin & Coolen (2011), Wiencierz (2013) and Schollmeyer (2021) for dealing with interval uncertainty within linear regression. If separate logistic regressions are generated via maximum likelihood estimation from the interval data and displayed as cumulative distribution functions, the envelope of the extreme functions bound the true model. The primary benefit of such an approach is that it represents the existing epistemic uncertainty removed by traditional methods. Additionally, this method can also handle the case of uncertainty in discrete risk factors. The imprecise probabilities approach makes the fewest assumptions, but some statistics can be computationally challenging for large datasets (Ferson et al., 2007).

In the case of uncertainty in the outcome status used within logistic regression, traditionally, there is little that can be done but to discard these data points as they cannot be used as part of the analysis. However, the proposed imprecise logistic regression technique can be used to include unlabeled examples within the dataset. This allows imprecise logistic regression to be extended to semi-supervised learning problems Amini & Gallinari (2002); Chi et al. (2019). Again the imprecise approach does not require making the smoothness, clustering and manifold assumptions that are usually required for semi-supervised learning (Chapelle et al., 2006).

## 2 Certain Logistic Regression

Let $\mathbf{x}$ be a $m$ dimensional covariate with a binary label $y \in \{1, 0\}$. Logistic regression can be used to model the probability that $y = 1$ using:

$$\Pr(y = 1 | \mathbf{x}) = \pi(\mathbf{x}) = \frac{1}{1 + \exp\left(-(\beta_0 + \beta_1 x_1 + \cdots + \beta_m x_m)\right)} \tag{1}$$

where $\beta_0, \beta_1, \ldots$ are a set of unknown regression coefficients. If there is a dataset $D$ with $n$ samples

$$D = \left\{ \begin{array}{c} \left((x_1^{(1)} x_2^{(1)} \cdots x_m^{(1)}), y_1\right) \\ \left((x_1^{(2)} x_2^{(2)} \cdots x_m^{(2)}), y_2\right) \\ \vdots \\ \left((x_1^{(n)} x_2^{(n)} \cdots x_m^{(n)}), y_n\right) \end{array} \right\} \tag{2}$$

the logistic regression model trained on $D$, $\mathcal{LR}(D)$, can be trained by finding optimal values of $\beta_0, \beta_1, \ldots, \beta_m$ for the observed data. This is often done using *maximum likelihood estimation* (Menard, 2010; Myung, 2003), although other techniques exist, for instance through Bayesian analysis Jaakkola (1997).

A classification, $\hat{y}$, can be made from the logistic regression model by assuming that

$$\hat{y} = \begin{cases} 1 & \text{if } \pi(\mathbf{x}) \geq C \\ 0 & \text{if } \pi(\mathbf{x}) < C \end{cases} \tag{3}$$

where $C$ is some threshold value. The simplest case is when $C = 0.5$ implying $\hat{y}$ is more likely to be true than false, however this value could be different depending on the use of the model and the risk appetite of the analyst. For example in medicine, a small threshold value may be used in order to produce a conservative classification and therefore reduce the number of false negative results. For the purpose of this paper where predictions are made with $C = 0.5$ unless otherwise stated.

## 2.1 Testing the Performance of the Logistic Regression

A synthetic 1-dimensional dataset ($D$) with a sample size of fifty was used to train a logistic regression model ($\mathcal{LR}(D)$); this model is shown in Figure 1. After training, it is useful to ask the question "how good is the model?" For logistic regression there are several ways in which that can be done, see Hosmer Jr et al. (2013, pp. 157–169) or Kleinbaum & Klein (2010, pp.318–326). For the analysis in this paper, we will consider (i) the sensitivity and specificity of the predictions made by the algorithm, (ii) receiver operating characteristic graph and area under curve statistic and (iii) discriminatory performance visualisations (Royston & Altman, 2010).

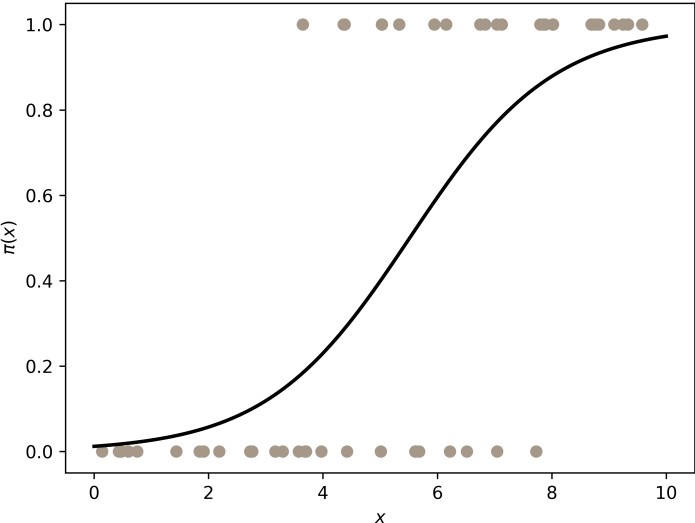

Figure 1: Logistic regression curve for the points shown.

We can make and compare the predictions made from the logistic regression model using a larger dataset that has been generated using the same method described above. Tabulating these results in a confusion matrix for the base predictions gives the following confusion matrix shown in Table 1. Two statistics are often used n order to express the performance of a classifier. These are the sensitivity $s$, the fraction of positive individuals correctly identified as such and the specificity $t$, the fraction of negative individuals correctly identified. Mathematically

$$s = \frac{True\ Positive}{Total\ Number\ of\ Positives} \tag{4}$$

and

$$t = \frac{True\ Negative}{Total\ Number\ of\ Negatives}. \tag{5}$$

|            | 1  | 0  | Total |
|------------|----|----|-------|
| Predicted 1 | 36 | 5  | 41    |
| Predicted 0 | 10 | 49 | 59    |
| Total      | 46 | 54 | 100   |

Table 1: Confusion matrix for 100 test data points using predictions from $\mathcal{LR}(D)$.

From Table 1 we can calculate that $s = 0.783$ and $t = 0.907$, which represent a good classifier. As confusion matrices and statistics are calculated from them depending on the cutoff value chosen ($C$ from Equation 3), a more complete way of determining the classification performance of models is by considering the receiver operating characteristic (ROC) curve of the model (Kleinbaum & Klein, 2010; Hosmer Jr et al., 2013). The ROC can be plotted by calculating how the sensitivity and specificity change for various threshold values and then plotting a graph of the false positive rate, $fpr = 1 - t$, against $s$ for all $C$ values. For the example, the ROC curve for $\mathcal{LR}(D)$ is shown in Figure 2a. The more upper-left a curve is, the better the classification. The worst performing model's ROC curve would match the black dotted line ($s = fpr$), corresponding to the ROC curve for a random classifier. If a model had a ROC curve down-right of this line, then it implies that the performance would be improved by switching the outcome classes of the model, as if it predicted true, then it is more likely to be false and vice versa. ROC curves can be compared graphically and by considering the area under the curve (AUC). The better the model is, the closer the AUC would be to 1. The worst possible AUC would be 0.5, as again, anything lower than that would be improved by simply switching the classification. For the ROC curve shown in Figure 2a $AUC = 0.917$, which could be considered 'outstanding discrimination' between the two classes (Hosmer Jr et al., 2013, p. 177).

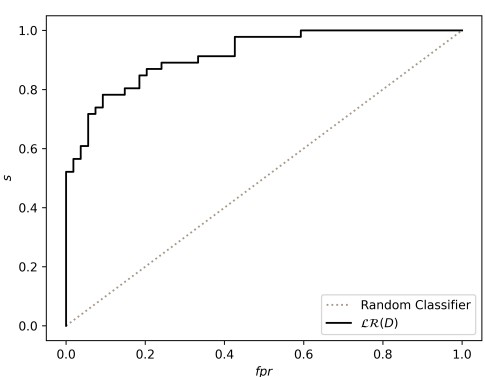

(a) ROC curve for the simple example.

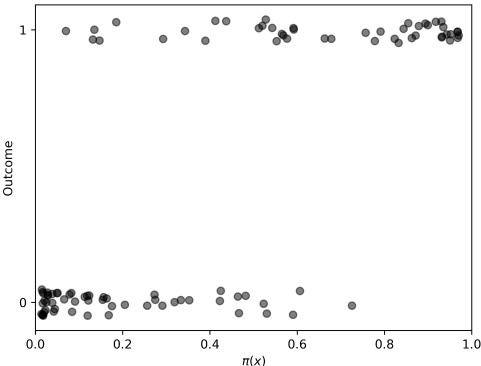

(b) Scatter plot of jittered outcome vs estimated probability for the simple example.

Figure 2: Two plots to show the discriminatory performance of the simple example.

Royston & Altman (2010) introduced visualisations to assess the discriminatory performance of the model by considering a scatter plot of the true outcome (jittered for clarity) vs the estimated probability. Such a plot is shown in Figure 2b. A perfectly discriminating model would have two singularities with all the points with outcome = 1 at (1,1) and all the points with outcome = 0 at (0,0). In general the better the classifier, the more clustered the points would be towards these values with the points on the upper band having larger probabilities and the points on the lower band having lower probabilities. From Figure 2b we can see that there is significant clustering towards the end points, showing that the model has excellent discriminatory performance.

## 3 Interval Uncertainty in Features

If there is interval uncertainty with the dataset

$$
D = \left\{ \begin{array}{c} \left( \left( \left[ \underline{x_1^{(1)}}, \overline{x_1^{(1)}} \right] \left[ \underline{x_2^{(1)}}, \overline{x_2^{(1)}} \right] \cdots \left[ \underline{x_m^{(1)}}, \overline{x_m^{(1)}} \right] \right), y_1 \right) \\ \left( \left( \left[ \underline{x_1^{(2)}}, \overline{x_1^{(2)}} \right] \left[ \underline{x_2^{(2)}}, \overline{x_2^{(2)}} \right] \cdots \left[ \underline{x_m^{(2)}}, \overline{x_m^{(2)}} \right] \right), y_2 \right) \\ \vdots \\ \left( \left( \left[ \underline{x_1^{(n)}}, \overline{x_1^{(n)}} \right] \left[ \underline{x_2^{(n)}}, \overline{x_2^{(n)}} \right] \cdots \left[ \underline{x_m^{(n)}}, \overline{x_m^{(n)}} \right] \right), y_n \right) \end{array} \right\} \tag{6}
$$

and we make no assumptions about the true value of $x_i^{\dagger(j)}$, only that $x_i^{\dagger(j)} \in \left[ \underline{x_i^{(j)}}, \overline{x_i^{(j)}} \right]$. Then it is only possible to partially identify an imprecise logsitic regression model for the data, $\overline{\mathcal{ILR}(D)}$:

$$
\mathcal{ILR}(D) = \left\{ \mathcal{LR}\left( D' \right) : \forall D' \in \left\{ \left\{ \begin{array}{c} \left( \left( x_1'^{(1)} \cdots x_1'^{(m)} \right), y_1 \right) \\ \vdots \\ \left( \left( x_n'^{(1)} \cdots x_n'^{(m)} \right), y_n \right) \end{array} \right\} \forall x_i'^{(j)} \in \left[ \underline{x_i^{(j)}}, \overline{x_i^{(j)}} \right] \right\} \right\} \tag{7}
$$

i.e. $\mathcal{ILR}(D)$ is the set off all possible logistic regression models that can be created from all possible datasets that can be constructed from the interval data, this ensures that the true logistic regression model, $\mathcal{LR}^{\dagger}$, is contained within the set. For continuous data this set is infinitely large. Predictions can be made by sampling all the possible models that are contained within the dataset and creating an interval containing the maximum and minimum values, $\left[ \underline{\pi(\mathbf{x})}, \overline{\pi(\mathbf{x})} \right]$.

As is it only useful to consider the minimum and maximum value of the predicted values then it is possible to reduce

$$
\mathcal{ILR}(D) = \left\{ \mathcal{LR}\left( D'_{\underline{\beta_i}} \right), \mathcal{LR}\left( D'_{\overline{\beta_i}} \right) \ \forall i = 0, 1, \ldots, m \right\} \\ \cup \left\{ \mathcal{LR}\left( \underline{D} \right), \mathcal{LR}\left( \overline{D} \right) \right\} \tag{8}
$$

where $D'_{\underline{\beta_0}}$ is the dataset constructed from points within the intervals so that the value of $\beta_0$ is minimized, $D'_{\overline{\beta_0}}$ is the dataset constructed from points within the intervals so that the value of $\beta_0$ is maximised, and so on. $\mathcal{LR}\left( D'_{\underline{\beta_0}} \right)$ can be found using mathematical optimisation. $\underline{D}$ corresponds to the dataset greated by taking the lower bound of every interval within $D$, similarly for $\overline{D}$. For a dataset with $m$ features there are $2 + 2m + 2^m$ values that are needed in order to find the bounds of the set.

When calculating the probability of a value being 1 under the imprecise model, as described above there is an interval probability $\left[ \underline{\pi(\mathbf{x})}, \overline{\pi(\mathbf{x})} \right]$. As such when using the model to perform classifications, this interval means that Equation 3 becomes

$$
\hat{y} = \begin{cases} 1 & \text{if } \underline{\pi}(\mathbf{x}) > C \\ 0 & \text{if } \overline{\pi}(\mathbf{x}) < C \\ [0,1] & \text{if } \pi(\mathbf{x}) \ni C \end{cases} \tag{9}
$$

The final line of this equation returns the *dunno* interval, meaning there is uncertainty in determining whether the datum should be predicted true or false. It is left up to the analyst to decide what should be done with such a result. However, it may be the case that if a prediction cannot be made then it may be useful to simply not make a prediction using logistic regression. Under this framework the traditional confusion matrix has an additional row as shown in Table 2. From this confusion matrix there are some useful statistics that can be calculated to account for the uncertainty produced by these uncertain classifications. The first is to

consider that the traditional definitions of sensitivity and specificity can be could be re-imagined by defining what the *predictive sensitivity s'* as the sensitivity out of the points for which a prediction was made

$$s' = \frac{a}{a+c} \tag{10}$$

and similarly the *predictive specificity t'* as the specificity for which a prediction was made

$$t' = \frac{d}{b+d}. \tag{11}$$

|  | 1 | 0 | Total |
|---|---|---|---|
| Predicted 1 | $a$ | $b$ | $P_+$ |
| Predicted 0 | $c$ | $d$ | $P_-$ |
| No Prediction | $e$ | $f$ | $P_x$ |
| Total | $T_+$ | $T_-$ | $N$ |

Table 2: Alternative confusions matrix where uncertain predictions are tabulated separately.

Two other statistics are useful to describe the data in Table 2. We can define the *positive incertitude $\sigma$* to be the fraction of positive cases for which the model could not make a prediction

$$\sigma = \frac{e}{a+c+e}. \tag{12}$$

Similarly, the *negative incertitude $\tau$* can be defined as the total number of negative cases for which the model could not make a prediction

$$\tau = \frac{f}{b+d+f}. \tag{13}$$

### 3.1 Example

Dataset $D$ from 2.1 has been intervalised into dataset $E$ using the following transformation $x \to [m - \epsilon, m + \epsilon]$ where $m$ is a number drawn from the uniform distribution $U(x-\epsilon, x+\epsilon)$ and $\epsilon = 0.375$ for all $x \in D$. Figure 3 shows the imprecise logistic regression curve. It evident that the interval datapoints have added considerable uncertainty to the regression. For comparison, a logistic regression model has been fitted on $E_m$ which is the dataset that has been deintervalised by taking the midpoint for all intervals in $E$ and the model from the original $D$ dataset from Figure 1.

As mentioned above, while making predictions from the imprecise model, one is likely to obtain dunno ranges as shown in Equation 9. We can tabulate the dunno intervals in the confusion matrix, giving the result shown in Table 3, from which the sensitivity and specificity are calculated as $s = [0.565, 0.783]$ and $t = [0.852, 0.963]$. Alternatively, we can tabulate the dunno predictions separately from the confusion matrix as has been done in Table 4. For observations for which the model produces a prediction, the predictive sensitivity is $s' = 0.722$ and the predictive specificity is $t' = 0.958$. The midpoint model, $\mathcal{LR}(E_m)$, has $s = 0.717$ and $T = 0.926$ both of which are lower than the $s'$ and $t'$. Analysts might devise alternative strategies to make a classification for the samples that were not given a prediction, potentially allowing improved outcomes. The incertitudes of the imprecise model are $\sigma = 0.217$, $\tau = 0.111$.

|  | 1 | 0 | Total |
|---|---|---|---|
| Predicted 1 | [26,36] | [2,8] | [28,44] |
| Predicted 0 | [10,20] | [46,52] | [56,72] |
| Total | 46 | 54 | 100 |

Table 3: Confusion matrix for 100 samples from the imprecise logistic regression model shown in Figure 3, tabulating inconclusive results as dunno intervals.

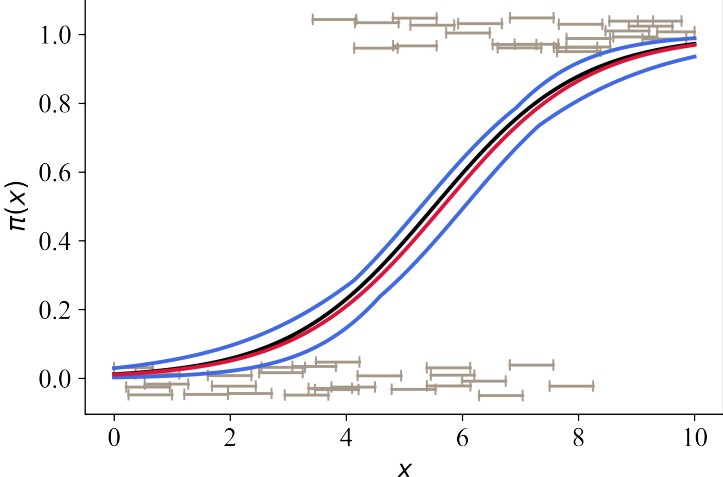

Figure 3: Imprecise logistic regression model (blue lines) for the interval data (grey, jittered for clarity), compared with the $\mathcal{LR}(D)$ (black line) Figure 1 and the model trained using the midpoints of the intervals (red line).

|  | 1 | 0 | Total |
|---|---|---|---|
| Predicted 1 | 26 | 2 | 28 |
| Predicted 0 | 10 | 46 | 56 |
| No Prediction | 10 | 6 | 16 |
| Total | 46 | 54 | 100 |

Table 4: Confusion matrix for 100 samples from the imprecise logistic regression model shown in Figure 3, tabulating inconclusive results separately.

The discriminatory performance of the model as a classifier can be assessed using visualisations, as shown in Figure 4. The simplest of these is the scatter plot shown in Figure 4a. We can see that all three models have good discrimination. We can also construct ROC plots and calculate the AUC for the ignored uncertainty and imprecise models. These plots are shown in Figure 4b. There are a few notable things about these plots; firstly, the model where the uncertain training data has been reduced to the midpoints is near-identical to that of the base model with $AUC = 0.9234$. For the imprecise model, the ROC is again bounded with an interval AUC of $[0.844, 0.956]$. We can also plot a ROC curve with $s'$ and $fpr'$ with $AUC = 0.942$.

It is also possible to consider situations where the data has been censored in some biased way. In Figure 5a the data has been biased by taking $x \to [x, x + 2\epsilon]$, setting the true value as always the lower bound of the interval. In Figure 5b the reverse has been done $x \to [x - 2\epsilon, x]$, setting the true value as always the upper bound of the interval. Finally, in Figure 5c the data has been intervalised using

$$ x \to \begin{cases} [x - 2\epsilon, x], & x < 5 \\ [x, x + 2\epsilon], & x \geq 5 \end{cases} $$

Looking at all these figures, we see that the imprecise model always bounds the base model. As a result, any interval regression analysis that has been performed is guaranteed to bound the true model, whereas there can be significant differences between the base model and the midpoint model.

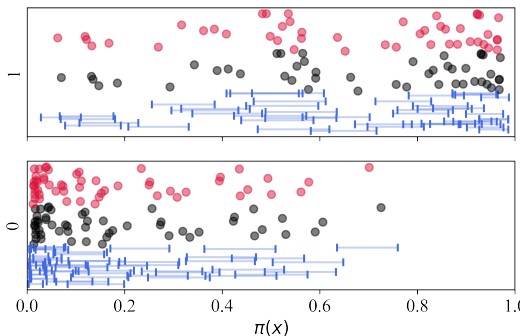

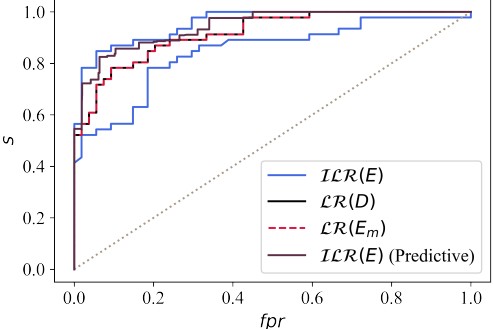

(a) Scatter plots of probability vs outcome for the base model (black), the model where uncertain datapoints have been excluded (red) and the imprecise model including the uncertain data (blue). The two outcomes have been separated into different plots for clarity.

(b) Receiver operating characteristic curve for the simple example with added uncertain classifications.

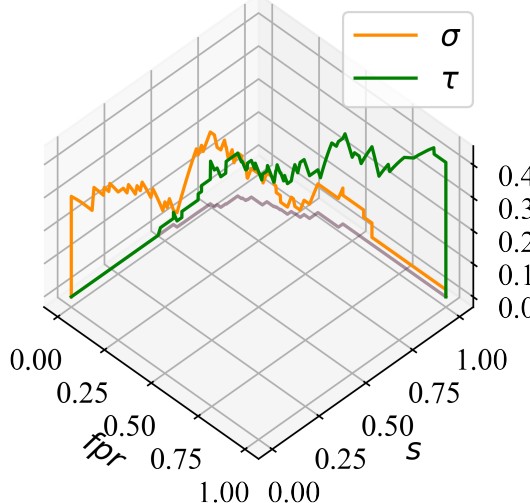

(c) 3 dimensional ROC curve for the simple example with positive/negative incertitude on $z$-axis. The blue line represents the 2-dimensional ROC curve shown in Figure 4b. The orange and green lines always lie above this blue line.

Figure 4: Three plots to show the discriminatory performance of the logistic regression models used within the simple example with uncertain classifications.

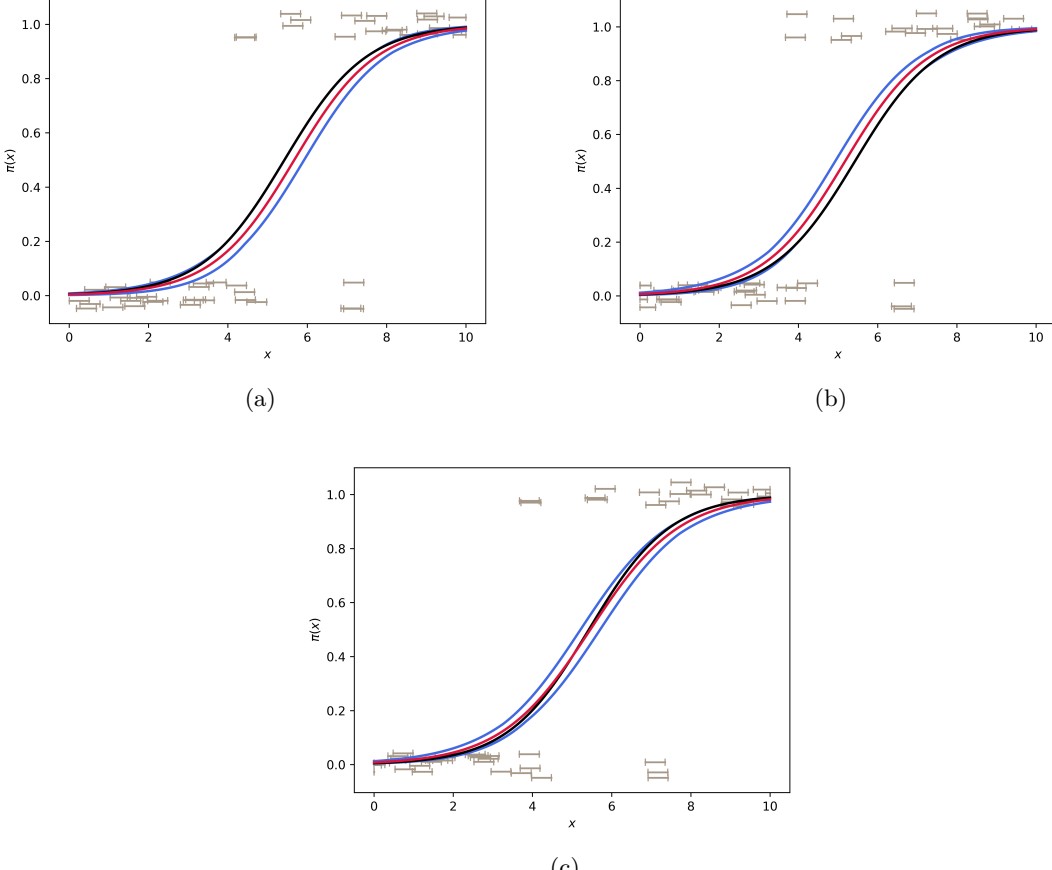

Figure 5: Logistic regression plots for interval data where the data has been intervalised in some biased way. In all the plots the blue bounds represent the imprecise logistic regression trained on the interval data, the red line represents the logistic regression trained by taking the midpoints of the interval and the black line is the logistic regression model trained on the base data as in Figure 1

## 4 Uncertainty in Labels

This set based approach can be extended to situation where there is uncertainty about the outcome status meaning there are some points for which we don't know the binary classification and can be represented as the dunno interval $[0, 1]$. In this situation the dataset $\mathcal{D}$ contains $p$ variables with corresponding labels $(\mathbf{x}_1, y_1), (\mathbf{x}_2, y_2), \cdots, (\mathbf{x}_p, y_p)$ but also $q$ variables for which the label is unknown $(\mathbf{x}_{p+1},\ ), (\mathbf{x}_{p+2},\ ), \cdots, (\mathbf{x}_{p+q},\ )$. Traditional analysis may just ignore these points. However they can be included within the analysis by considering the set of possible logistic regression models trained on all possible datasets that could be possible based upon the uncertainty. This set of datasets can be created by giving all unlabeled values the value 0, all unlabeled values the value 1 and all combinations thereof, i.e.

$$\mathcal{ILR} = \left\{ \mathcal{LR}(D') \; \forall D' \in \left\{ \begin{array}{c} \{(\mathbf{x}_1, y_1), \cdots, (\mathbf{x}_p, y_p), (\mathbf{x}_{p+1}, 0), \cdots, (\mathbf{x}_{p+q}, 0)\} \\ \{(\mathbf{x}_1, y_1), \cdots, (\mathbf{x}_p, y_p), (\mathbf{x}_{p+1}, 0), \cdots, (\mathbf{x}_{p+q}, 1)\} \\ \vdots \\ \{(\mathbf{x}_1, y_1), \cdots, (\mathbf{x}_p, y_p), (\mathbf{x}_{p+1}, 1), \cdots, (\mathbf{x}_{p+q}, 0)\} \\ \{(\mathbf{x}_1, y_1), \cdots, (\mathbf{x}_p, y_p), (\mathbf{x}_{p+1}, 1), \cdots, (\mathbf{x}_{p+q}, 1)\} \end{array} \right\} \right\} \tag{14}$$

This leads to $2^q$ possible logistic regression models. An imprecise logistic regression model can then be created by finding the envelope of the set. As the computational time for this algorithm increases as $\mathcal{O}(2^q)$, then as $q$ increases finding the bounds by calculating the envelope for all possible combinations can become computationally expensive. We can reduce the complexity by again finding $\mathcal{LR}\left(D'_{\underline{\beta_i}}\right),\mathcal{LR}\left(D'_{\overline{\beta_i}}\right)$, etc through optimization.

## 4.1 Example

Dataset $F$ has been created from dataset $D$ by replacong 5 labels from the dataset with the dunno interval. The labels that have been changed are around the point at This set-based approach can be extended to situations where there is uncertainty about the outcome status meaning there are some points for which we do not know the binary classification and can be represented as the dunno interval $[0,1]$. In this situation the dataset $\mathcal{D}$ contains $p$ variables with corresponding labels $(\mathbf{x}_1,y_1),(\mathbf{x}_2,y_2),\cdots,(\mathbf{x}_p,y_p)$ but also $q$ variables for which the label is unknown $(\mathbf{x}_{p+1}, ),(\mathbf{x}_{p+2}, ),\cdots,(\mathbf{x}_{p+q}, )$. Traditional analysis may ignore these points. However, they can be included within the analysis by considering the set of possible logistic regression models trained on all possible datasets that could be possible based upon the uncertainty. This set of datasets can be created by giving all unlabeled values the value 0, all unlabeled values the value 1 and all combinations thereof, i.e.

$$\mathcal{ILR} = \left\{ \mathcal{LR}(D') \; \forall D' \in \left\{ \begin{array}{c} \{(\mathbf{x}_1,y_1),\cdots,(\mathbf{x}_p,y_p),(\mathbf{x}_{p+1},0),\cdots,(\mathbf{x}_{p+q},0)\} \\ \{(\mathbf{x}_1,y_1),\cdots,(\mathbf{x}_p,y_p),(\mathbf{x}_{p+1},0),\cdots,(\mathbf{x}_{p+q},1)\} \\ \vdots \\ \{(\mathbf{x}_1,y_1),\cdots,(\mathbf{x}_p,y_p),(\mathbf{x}_{p+1},1),\cdots,(\mathbf{x}_{p+q},0)\} \\ \{(\mathbf{x}_1,y_1),\cdots,(\mathbf{x}_p,y_p),(\mathbf{x}_{p+1},1),\cdots,(\mathbf{x}_{p+q},1)\} \end{array} \right\} \right\} \tag{15}$$

This leads to $2^q$ possible logistic regression models. An imprecise logistic regression model can then be created by finding the envelope of the set. As the computational time for this algorithm increases as $\mathcal{O}(2^q)$, then as $q$ increases finding the bounds by calculating the envelope for all possible combinations can become computationally expensive. We can reduce the complexity by again finding $\mathcal{LR}\left(D'_{\underline{\beta_i}}\right),\mathcal{LR}\left(D'_{\overline{\beta_i}}\right)$, etc through optimization.

## 4.2 Example

Dataset $F$ has been created from dataset $D$ by replacing five labels from the dataset with the $[0,1]$ interval. The labels that have been changed are around the point at which the data goes from 0 to 1. $\mathcal{IRL}(F)$ is the imprecise logistic regression model that is trained on this uncertain dataset and is shown in Figure 6. For comparison, $\mathcal{LR}(F_\times)$ is the model trained on the dataset with the dunno labels removed and $\mathcal{LR}(D)$ is also shown. From the figure, it is clear that the uncertain labels add significant uncertainty to the model. It can also be seen that, as expected, the imprecise model bounds the 'true' model.

When it comes to assessing classifications from the model, there are two possible ways of expressing the effect of uncertainty presented by the imprecise model. One approach is to consider the prediction as the dunno interval $[0,1]$ within a traditional confusion matrix. Doing this we get the confusion matrix shown in Table 5, from which the sensitivity and specificity can be calculated as intervals $s = [0.717,0.848]$ and $t = [0.796,0.944]$. The other method is to tabulate the dunno results separately within the confusion matrix as has been done in Table 6. From this we can clearly see that for observations for which the model produces a prediction it performs well, $s' = 0.825$ and $t' = 0.935$. This shows that, for the predictions that the model did make, there were fewer mistakes than when the analysis ignored the uncertain data points, although this has come at the cost of having a few data points for which no classification was made. For the incertitude we find $\sigma = 0.130$ and $\tau = 0.148$.

As before, it is useful to consider visualisations when discussing the discriminatory performance of the classifier (Figure 7). The simplest of these are the scatter plots shown in Figure 7a. We can see that all the models have good discrimination. We can also construct ROC plots and calculate their AUCs. The ROC

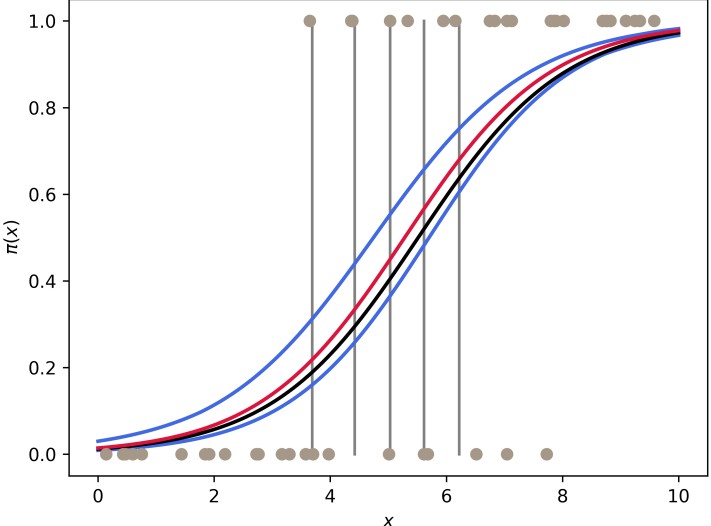

Figure 6: Bounds for the imprecise logistic regression (blue) for all the 50 grey points with 5 points made uncertain (shown with vertical lines with the true values shown with black diamonds). Compared with $\mathcal{LR}(D)$ from Figure 1 (black) and $\mathcal{LR}(F_\times)$ trained on the dataset removing the uncertain points (red).

|  | 1 | 0 | Total |
|---|---|---|---|
| Predicted 1 | [33,39] | [3,11] | [36,50] |
| Predicted 0 | [7,13] | [43,51] | [50,64] |
| Total | 46 | 54 | 100 |

Table 5: Confusion matrix for 100 datapoints from the imprecise logistic regression model shown in Figure 6, keeping uncertain predictions as the interval $[0, 1]$.

plots are shown in Figure 7b. The $\mathcal{ILR}(F)$ model has $AUC = [0.847, 0.960]$, the no prediction model has $AUC = 0.946$ and $\mathcal{LR}(F_\times)$ again aligns with $\mathcal{LR}(D)$ with $AUC = 0.917$. It is also worth considering how the incertitude changes as the threshold value changes. We can plot a 3-dimensional version of the ROC plot, with the positive/negative incertitude on the $z$-axis. Such a plot is shown in Figure 7c. From this plot, we can see that as the sensitivity improves, the positive incertitude generally decreases and as the specificity increases, the negative incertitude decreases.

## 5 Interval Uncertainty in Predictor Variables and Outcome Status

The imprecise approach can be used when there is uncertainty about both the features and the labels. Such situations are present in numerous real-world datasets. For example, Osler et al. (2010) use a logistic regression model to predict the probability of death for a patient after a burn injury. The model that they use is based upon a subset of data from the American Burn Association's National Burn Database[1]. The dataset has a mix of discrete (gender, race, flame involved in injury, inhalation injury) and continuous variables (age, percentage burn surface area) that can be used to model the probability that a person dies (outcome 1) after suffering a burn injury. Osler et al. exclude some patients from the dataset before training their model. They remove patients if their age or 'presence of inhalation injury' was not recorded. Additionally, as patients older than 89 years were assigned to a single age category in the original dataset, they gave them a random age between 90 and 100 years.

---

[1] http://ameriburn.org/research/burn-dataset/

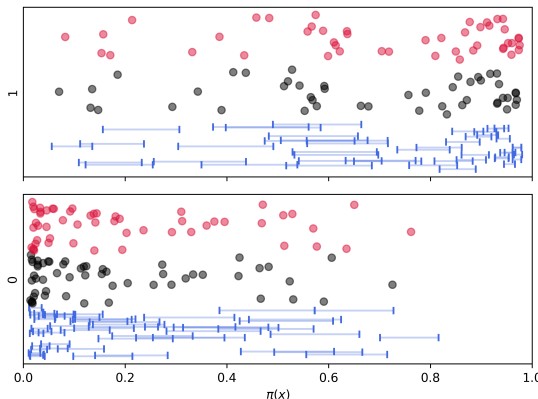

(a) Scatter plots of probability vs outcome for $\mathcal{LR}(D)$ (black), $\mathcal{LR}(F_\times)$ (red) and $\mathcal{ILR}(F)$ (blue). The two outcomes have been separated into different plots for clarity.

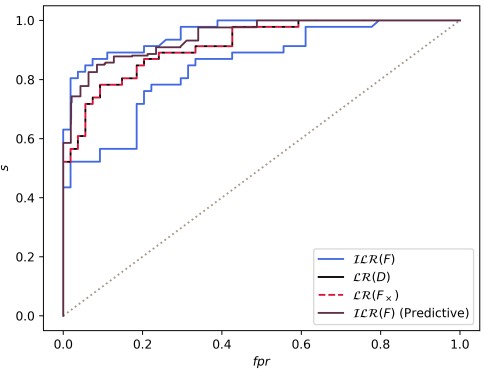

(b) Receiver operating characteristic curve for the uncertain example with added uncertain classifications.

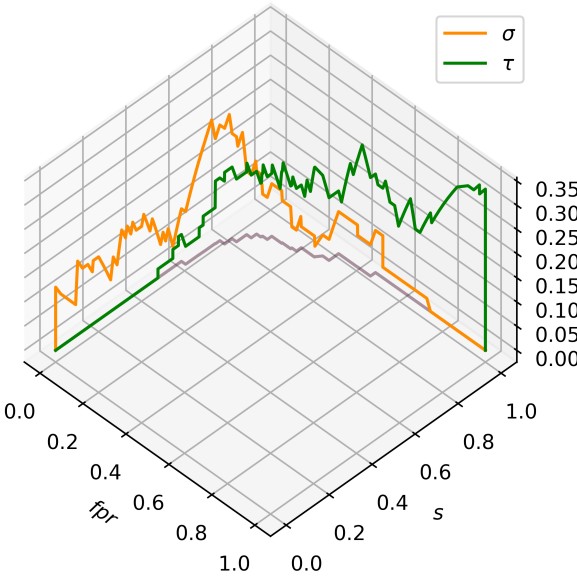

(c) 3 dimensional ROC curve for the uncertain label example with positive/negative incertitude on $z$-axis. The blue line represents the 2-dimensional ROC curve shown in Figure 7b. The orange and green lines always lie above this blue line.

Figure 7: Three plots to show the discriminatory performance of the logistics regression models used within the simple example with uncertain labels.

|              | 1  | 0  | Total |
|--------------|----|----|-------|
| Predicted 1  | 33 | 3  | 36    |
| Predicted 0  | 7  | 43 | 50    |
| No Prediction | 6  | 8  | 14    |
| Total        | 46 | 54 | 100   |

Table 6: Confusion matrix for 100 datapoints from the imprecise logistic regression model shown in Figure 6, tabulating uncertain predictions separately.

Osler et al. did not need to exclude these patients merely because of epistemic uncertainty about the values. The proposed approach can be used with the original data. For instance, patients for which the outcome was unknown could have been included within their analysis as described in Section 4. Similarly, patients for which inhalation injury or age was unknown could have been included with the method described in Section 3. Patients with unknown inhalation injury could have been included as the $[0, 1]$ interval. Patients whose age was completely unknown could have been replaced by an interval between the minimum and maximum age, whereas if there was uncertainty because they were over 90 years old, then they could be intervalised as $[90, 100]$.

Other interval uncertainties may be present within the dataset. It is unlikely to be the case that all the people used within the study fit neatly into the discrete variables given. For instance the variable race is valued at 0 for "non-whites" and 1 for "whites". However, it goes without saying that the diversity of humanity does not simply fall into such overly simplified categories; there are likely to be many people who could not be given a value of 0 or 1 and should instead have a $[0, 1]$ value. The same is true for gender. Not everyone can be defined as male or female. Also, there is almost certainly some measurement uncertainty associated with calculating the surface area of the burn that may also be best expressed as intervals. For simplicity, these uncertainties have not been addressed below.

For use in this analysis, the subsample of the dataset used by Osler et al. that was made available by Hosmer Jr et al. (2013, p. 27) has been used. This version of the dataset includes 1000 patients from the 40,000 within the full study and has a much higher prevalence of death than the original dataset. Because access to the original data is prohibitively expensive, the values in this dataset have been reintervalised to replicate some of the removed uncertainty to create a hypothetical dataset, $B$, for this exposition. As there are no individuals older than 90 within the dataset, that particular reintervalisation has not been possible, so all patients who were older than 80 have had their ages intervalised as [80,90]. Similarly, for 20 patients, the censored inhalation injury has been restored to dunno interval. Ten patients, who had been dropped because their outcome status was unknown, have been restored with status represented as [0,1].

There are two possible routes in which an analyst could proceed when faced with such a dataset. They could follow the original methodology of Osler et al. and randomly assign patients with interval ages a precise value and then discard all other patients for which there is some uncertainty. Alternatively, the analyst could include the uncertainty within the model by creating an imprecise logistic regression model. As there is uncertainty within both the features and the labels, the model can be estimated by finding the values within the intervals that correspond to the minimum and maximum for $\beta_0$, $\beta_1$, etc. $\mathcal{ILR}(B)$ is the imprecise logistic regression fitted from this burn data. For comparison, $\mathcal{LR}(B_\times)$ has also been fitted based on removing the uncertainty in $B$ using the same methodology as Osler.

When it comes to the performance of the two models, we can again turn to visualisations, as shown in Figure 8. Firstly, looking at Figure 8c we can see that the vast majority of patients who were given a low probability of death ($\pi$) did indeed survive, and patients who were given a high probability of death did sadly die. The ROC plots are shown in Figure 8a, Figure 8b shows the upper right corner of the plot in more detail. $\mathcal{ILR}(B)$ has a $AUC = [0.955, 0.974]$, the no prediction model has $AUC = 0.972$ and $\mathcal{LR}(B)$ has $AUC = 0.966$.

It is pertinent to consider how a model is likely to be used and how uncertainty about the predicted probability of death impacts the classification. One method of dealing with this uncertainty that arises in Sections 4 and 3 is simply not making a prediction when the interval for $\pi$ straddles $C$. This method may not be appropriate

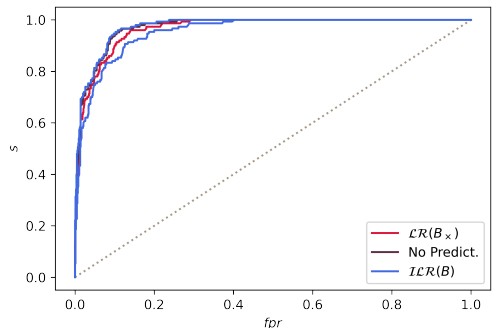

(a) Receiver operating characteristic curves for the burn example.

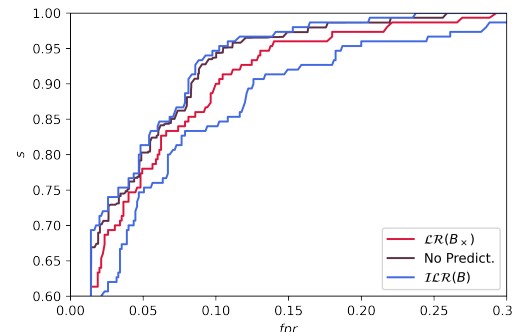

(b) Receiver operating characteristic curves for the burn example.

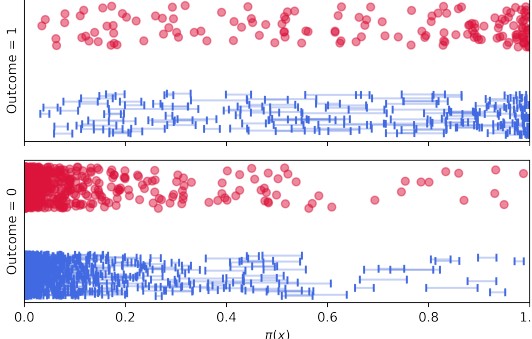

(c) Scatter plots of probability vs outcome for the model where uncertain values have been excluded (red) and the model including the uncertain values (blue). The two outcomes have been separated into different plots for clarity.

Figure 8: Plots to show the discriminatory performance of the various logistic regression models for the burn survivability example

in this example. What should happen when the model is unable to make a prediction should depend on what the result of deciding a patient has a high risk of death means clinically. If the model was being used to triage patients that need to go to a major trauma centre because the probability of death is considered high, then–out of an abundance of caution–one might prefer that if any part of the interval probability was greater than some threshold, the patient should be considered high risk. Equivalent To taking the probabilities from the upper bound of the range,

$$
\text{high risk} = \begin{cases} 1, & \text{if } \overline{\pi} \geq C \\ 0, & \text{otherwise.} \end{cases} \tag{16}
$$

However, if patients who are considered high risk then undergo some life-altering treatment that is perhaps only preferable to death, then under the foundational medical aphorism of "first do no harm", it may be preferable to consider a patient high risk only if the whole interval is greater than the decision threshold, this is equivalent of taking the probabilities from the lower bound of the range,

$$
\text{high risk} = \begin{cases} 1, & \text{if } \underline{\pi} \geq C \\ 0, & \text{otherwise.} \end{cases} \tag{17}
$$

## 6 Discussion

Many uncertainties are naturally expressed as intervals, and it is better to compute with what we know than to make assumptions that may need to be revised later. In the case of logistic regression, when faced with interval uncertainties, samples are often dropped from analyses–assuming that they are missing at random– or reduced down to a single value. In this paper, we have shown that this need not be the case. Interval uncertainties can be included within a logistic regression model by considering the set of possible regression models as an imprecise structure. This even includes situations where there is uncertainty about the outcome status. It is not reasonable to throw away data when the status is unknown if the reason the data has gone missing is dependent on the value or status of the missing samples. It is unlikely to be the case that all the predictor values used within the studies are often associated with non-negligible interval uncertainties. This uncertainty should not simply be thrown away because it makes the subsequent calculations easier.

When using an imprecise model, each new sample gets an interval probability of belonging to one of the binary classifications. When it comes to making classifications from the model, there are likely to be samples for which a definitive prediction cannot be made. If one is happy to accept a *don't know* result, then the deterministic performance can be improved for the samples for which a prediction could be made. It may seem counterproductive or unhelpful for a model to return a don't know result. However, this can be desirable behaviour; saying "I don't know" is perfectly valid in situations where the uncertainty is large enough that a different decision could have been reached. Uncertainty in the output can allow for decisions made by algorithms to be more humane by requiring further interrogation to make a classification. Alternatively, depending on the use case, other ways of making decisions based on uncertain predictions could be made.

This paper used crude algorithms to compute the imprecise logistic regression, and no guarantee of coverage is made. As this approach is NP-hard, future work in this area should be invested in finding improved algorithms to make them practical for large-scale datasets and guarantee coverage.

To conclude, we have shown that it is possible to include uncertainty in both outcome status and predictor variables within logistic regression analysis by considering the set of possible models as an imprecise structure. Such a method can clearly express the epistemic uncertainties within the dataset that are removed by traditional methods.

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
