# OpenReview forum: "Logistic Regression Through the Veil of Imprecise Data"
_TMLR — Withdrawn by Authors_

### Review · Reviewer_NWvy · 2022-07-06

**Summary Of Contributions:**

        This paper describes a method to incorporate input and output uncertainty in the context of logistic regression. Uncertainty in the inputs is considered to be specified by some intervals in which the input features can take values. The output uncertainty is specified simply by incorporating all potential data instances with all potential label combinations. The method proposed is evaluated on some problems and their benefits are shown.


**Requested Changes:**


It is not clear how the proposed method works and is trained when there is uncertainty in the inputs and in the outputs. The authors have to make a better effort explaining their method. The experiments are also weak. It seems that only a few datasets have been considered. The authors should test their method on a wider range of problem, both for input uncertainty and also label uncertainty. The paper is missing comparisons with related methods. The authors explain in the introduction that there are other methods to deal with these problems in logistic regression. However, it seems the do not compare results with them.  See, e.g.,

(Bertrand, 2000; Billard & Diday, 2000; Bock & Diday, 2001; De Souza et al., 2011)

Broader Impact Concerns:


**Strengths And Weaknesses:**

Strengths:

        - The proposed method seems to be simple.

Weaknesses:

        - The cost of the method seems high.

        - The proposed method is not very clearly explained. It is not clear how the authors consider all combinations of data instances compatible with the hypothesis made.

        - The experimental section is weak since there are no comparisons with other methods and the authors only consider a few datasets.

---

### Review · Reviewer_CtMB · 2022-07-07

**Summary Of Contributions:**

The paper explores the probabilistic nature of classifiers, namely, of the logistic regression. It is proposed to estimate a number of logistic regression models to construct interval uncertainties.


**Broader Impact Concerns:**

No ethical issues.

Potentially, the proposed approach can have advantages of probabilistic and interpretable classifiers.

**Requested Changes:**

I am not sure that the paper has enough both technical and experimental novelty to be published in its current form.

It is stated that "the better the classifier, the more clustered points would be towards these values..." (Page 4): I am not sure, since if a classifier is not well calibrated, the probabilities can be peaked but not necessarily correspond to the true conditional probabilities.

Equation 8: it would be more clear if it is somewhere explicitly defined that $\pi = [\barbelow{pi} \bar{pi}]$.

Figure 4 is not clear. What exactly it illustrates? Especially, 4c) ?

Section 4.1 is repeated twice.


**Strengths And Weaknesses:**

Strengths. The method is described in details. The evaluation metrics are also well explained.


Weaknesses. The description and explanations of the results are quite basic and straightforward. So, the description of the ROC is very detailed (as well as the description of the logistic regression), and some parts of the paper resemble more a textbook than a scientific paper. It is also stated that the results presented in Table 1 (confusion matrix) represent a good classifier, however, good/bad performance depends on an application, and an error rate on arbitrarily simulated data does not allow to do such strong conclusions.

In Section 3, there is a problem I guess: "the value of \beta_0 is minimised", "\beta_0 is maximised": the goal is to minimise (or maximise) the likelihood function and not the parameter values.

Section 4 introduces a very complex model with an exponential number of logistic regressions, and the motivation for such a complex model is lacking.

It is difficult to deduce what figures illustrate, e.g., Figure 6: Is the blue line (the proposed imprecise logistic regression) is a good model or not? What this interval means?

Comparisons with other methods are absent. It is not clear, when the proposed method should be used?
And, as far as I understood, there are not any guarantees for the proposed method.

---

### Review · Reviewer_2nfi · 2022-07-08

**Summary Of Contributions:**

This paper proposes a way to compute logistic regression when the observations are only partially known. Precisely, instead of having access to the true feature value, the learner is provided with an interval in which this value lies. Similarly, unknown labels can be modeled by the set {0, 1}. While previous methods usually consist in replacing the intervals by a single value (e.g., the mean, the median), the authors propose to learn as many Logistic Regressions as there are possibilities for the feature values (this means possibly infinitely many models in the case of intervals) and to output $\underline{\pi}(x)$ and $\bar{\pi}(x)$ the minimal and maximal probability of being 1 given input $x$ among these models. The key point is that only a finite number of models can be considered to compute  $\underline{\pi}(x)$ and $\bar{\pi}(x)$. The prediction is then 1 if  $\underline{\pi}(x) > C$ and 0 if $\bar{\pi}(x) < C$, given some threshold $C$, and _abstain_ if  $\bar{\pi}(x) \le C \le \underline{\pi}(x)$. Experiments, mainly on one dimensional synthetic data are provided to support the proposed approach.

**Broader Impact Concerns:**

I do not have any concerns on the ethical implications of this work.

**Requested Changes:**

- More explanations on Equation (8)

- More details about the "mathematical optimisation" that allows to find the datasets with the extremal coefficients

- More detailed experiments (fair comparison, more benchmarks)

- Polishing the writing

**Strengths And Weaknesses:**

Although the bias taken in this article, which consists in trying to leverage the entire interval, is interesting, I have some concerns.

- I might be missing something, but I cannot see why $\mathcal{ILR}(D)$ can be reduced to Equation (8). I agree that given some $x \in \mathbb{R}^m$, and some $\beta \in \mathbb{R}^m$ with $\beta_i \notin {\underline{\beta}_i, \bar{\beta}_i}$, we have $x^\top \beta' \ge x^\top \beta$ if $x_i$ is positive (respectively negative) and $\beta'$ is equal to $\beta$ except that the entry $i$ has been replaced by $\bar{\beta}_i$ (respectively $\underline{\beta}_i$). But do we know for sure that such a $\beta'$ is a solution to a Logistic Regression for some choice of dataset? I think this deserves more explanations.

- Although, the way the datasets yielding the extremal values for the coefficients $\beta_i$ are found should be better explained in my opinion, as it is a critical part of the proposed approach. In the present submission, it is only written the very vague "can be found using mathematical optimisation".

- On a minor note, Equation (8) seems to contain only $2 + 2m$ models while it is mentioned $2 + 2m + 2^m$ in the paragraph below. Where does this come from?

- I am also concerned with the way the method is compared to benchmarks. Indeed, ILR is by definition allowed to abstain on a certain number of points, and sensitivity and specificity are computed only on points for which ILR made a prediction. Although it is an advantage of the method to clearly provide with a criterion to abstain, I feel it would be fairer to not consider these points to evaluate midpoints LR. In particular I expect $\pi_{mid}(x)$ to be close to $1/2$ for these points.

- Another possibility could be to provide a random guess when $\bar{\pi}(x) \le C \le \underline{\pi}(x)$. But then it seems that we are left with the exact same problem we tried to solve initially: instead of simplifying intervals to points we kept intervals but produce interval predictions, that we have now to turn into single predictions. Can the authors provide arguments in favor of their approach w.r.t. this aspect?

- I feel that the experiments could be strengthened, by showing more benchmarks such as missing values imputations and considering data outside of 1D data

The authors should proofread their manuscript.
-  Section 4.1 does not make any sense. I don't think this is acceptable, especially for a journal venue with no deadline.
- "We can make and compare the predictions made"
- "often used n order to express"
- "which represent a good classiﬁer" (on which basis can you say this?)
- computed might be better than calculated
- the first sentence of Section 3 starts with "if" but doe not contain any verb
- I think $\underline{\pi}(x)$ and $\bar{\pi}(x)$ should be defined properly
- I don't agree with the "=" sign in Equation (8)
- "greated by"
- a, b, c, d are not defined in Equations (10), (11)
- "contains p" observations/samples maybe, rather than "variables"?

---

### Author Response · Authors · 2022-08-03
**Withdrawal of paper**

Dear Reviewers and Editors,

I am withdrawing my manuscript to make more substantial changes that will take longer than the decision process allows.

Thank you for the time spent reviewing my paper. I intend to resubmit once these changes have been made.

---

### Note · Authors · 2022-08-03

I have read and agree with the venue's withdrawal policy on behalf of myself and my co-authors.